# A Facile Way to Modify Polyester Fabric to Enhance the Adhesion Behavior to Rubber

**Hongwei He [1,2,\*], Pengfeng Wu [1,2], Zeguang Yang [1,2], Zhihao Shi [1,2], Wenwen Yu [1,2], Fuyong Liu [1,2], Fengbo Zhu [1,2], Qiang Zheng [1,3,\*], Dezhong Zhang [4,5] and Shumin Li [4,5]**

1   College of Materials Science & Engineering, Taiyuan University of Technology, Taiyuan 030024, China
2   Key Laboratory of Interface Science and Engineering in Advanced Materials, Ministry of Education, Taiyuan University of Technology, Taiyuan 030024, China
3   Ministry of Education Key Laboratory of Macromolecular Synthesis and Functionalization, Department of Polymer Science and Engineering, Zhejiang University, Hangzhou 310027, China
4   Huayang New Material Technology Group Co., Ltd., Yangquan 045000, China
5   YMC Navalley (Shanxi) Aerogel Science and Innovation City Management Co., Ltd., Yangquan 045000, China
\*   Correspondence: hehongwei4103@126.com (H.H.); zhengqiang@zju.edu.cn (Q.Z.)

**Abstract:** Due to the extremely inert surface of polyester (PET) fabric, a toxic and traditional resorcinol-formaldehyde-latex (RFL) dipping solution is always necessary in the rubber composite industry. Unfortunately, other effective methods for fabric surface treatment are in urgent need to improve the poor bonding interface between the fabric and the rubber matrix. In our study, a facile way to modify PET fabric was developed. Specifically, the fabric was treated by an alkaline solution and a coupling agent with magnetic agitation. Afterwards, the treated fabric/rubber composites were prepared through a co-vulcanization process. Attenuated total reflectance Fourier transform infrared spectroscopy (ATR-FTIR), and thermogravimetric analysis (TGA) were used to characterize the surface chemical composition of the modified fabrics. The adhesion behavior was analyzed by the peel test. The results showed that the fabric surface was successfully grafted with a coupling agent, and the peel strength reached 9.8 N/mm after KH550 treatment, which was an increase if 32% compared with that of the original fabric/rubber composite. In addition, the vulcanization rate and interfacial fracture mechanism are also researched.

**Keywords:** polyester fabric; surface; rubber; composites; adhesion behavior

## 1. Introduction

Polyester (PET) fabrics are widely applied to reinforce rubber compounds, such as in conveyor belts, v-belts, tires and high-pressure hoses. They are important to develop rubber-based industries [1]. However, the inert surface and modulus of the synthetic fibers have a poor interaction with rubber, and the corresponding fabric/rubber composites have poor wettability between the fabric and the rubber matrix [2]. These phenomena constrain the overall properties, such as safety and durability, which can be greatly changed due to fiber modification [3,4]. Two methods are used to combine fabric and rubber. For the first method, the fabric and unvulcanized rubber are placed in a steel mold and vulcanized [5,6]. In the second method, the fabric is merged with the vulcanized rubber by means of rubber adhesive [7]. Compared with the second method, the first method is simple and low cost because expensive rubber adhesives are avoided in the industrial applications.

As a synthetic fiber, polyester (PET) fibers are suitable for reinforcing rubber as skeleton materials because of their good mechanical properties and wear resistance. Physical or chemical treatment is intensively conducted to improve the adhesion between the PET fabric and the rubber matrix, such as plasma treatment, γ-ray irradiation, electron beam (EB) irradiation, the coating method, UV-initiated grafting, oxidative acid etching and chemical grafting [8,9]. However, plasma and γ-ray irradiation as physical methods are high cost

and need rigorous experimental equipment. Thus, surface grafting and etching, being more cost-effective, can meet the demand for high performance fabric/rubber composites [10].

Thus far, PET fabrics, as reinforcing materials, are usually coated with traditional resorcinol-formaldehyde-latex (RFL) dipping solution. However, PET fabrics have an inert surface and absence of active functional groups makes for unacceptable bonding between the fabric and the rubber [2]. In other words, the untreated PET fabric surface can be wet and this affects compatibility with RFL adhesive, because the carboxyl and hydroxyl of PET do not easily react with the hydroxyl and methylol groups of RFL.

In this work, we propose a facile way to modify polyester fabric to enhance the adhesion behavior to rubber as well as a co-vulcanization process. We studied the chemical structure and interfacial adhesion mechanism between fabric and rubber by Fourier transform infrared spectra (FTIR), thermogravimetric analysis (TGA) and peel test. By changing the fabric surface, the adhesion of PET fabrics to rubber could be significantly increased, It is a promising application in engineering.

## 2. Experimental

### 2.1. Materials

Styrene-butadiene rubber (SBR, ESBR 1502, styrene/butadiene (23/77 of mass ratio)) was supplied by Sinopec Qilu Co., Ltd., Zibo, China, and nature rubber (NR, 1-0082) was from Yunnan Agricultural Reclamation Group Co., Ltd., Kunming, China. Aromatic oil (VIVATEC 500) was supplied by Hansen & Rosenthal Group, Ningbo, China. Carbon black N330 was purchased from Kabote Investment Co., Ltd., Shanghai, China. The adhesive agents, RA and RS, were obtained from Jiangsu national Huagong Technology Co., Ltd., Nanjing, China. ZnO was supplied by Liuzhou Zinc Products Co., Ltd., Liuzhou, China. Stearic acid (SA) was supplied by Fengyi Oil Technology Co., Ltd., Zhengzhou, China. Accelerator was supplied by Qingdao Kangan Rubber Technical Co., Ltd., Qingdao, China. Silane coupling agent KH550 was obtained from Nanjing Chuangshi chemical additives Co., Ltd., Nanjing, China. NaOH was obtained from Sinopharm Chemical Reagent Co., Ltd., Shanghai, China. Anhydrous ethanol was obtained from Tianjin Tianli Chemical Reagent Co., Ltd., Tianjin, China.

### 2.2. Surface Treatment of the Fabric

The fabric was put into deionized water and anhydrous ethanol with ultrasonication for 2 h to clean up lubricant and water-soluble and fat-soluble impurities on the fiber. Then, the fabric was dried in an oven at 70 °C for 24 h. The fully dried fabric was soaked in 2 wt% NaOH solution for 2 h, where it hydrolyzed the surface of the PET fabric under magnetic agitation; it was labeled as NaOH-treated fabric. NaOH-treated fabric was soaked in the ethanol solution of 5 wt% silane coupling agent (KH550) with magnetic agitation for 2 h to alkylate, and was labeled as KH550-grafted fabric. Last, the fabric was dried in an oven at 70 °C for 24 h.

### 2.3. RFL Dipping Solution

Formaldehyde, resorcinol and sodium hydroxide, as a catalyst, were mixed in water and, then, the mixture was kept still and matured for 6 h at room temperature, forming matured resorcinol-formaldehyde (RF) resin [11]. Ammonium hydroxide (in order to remove free formaldehyde) and vinyl pyridin were added to the matured RF solution to prepare the RFL solution and kept still for 20 h at room temperature. The components of traditional RFL dipping solution are given in Supplementary Materials Table S1.

### 2.4. Preparation of PET/Rubber Composites

The preparation of PET/rubber composites, shown in Figure 1, was as follows.. The composition of the rubber compound, shown in Supplementary Materials Table S2, was used in our study. The weighed rubber compound was put into a two-roll mill, in which rubber sheets, with a thickness of 2 mm, were obtained by adjusting the space of the

two-roll mill. Then, the rubber sheets were cut into the same size as the fabric, $25 \times 25$ cm$^2$. Finally, the fabric was carefully laid between the two rubber sheets to form a sandwich structure composed of 1 layer of fabric between2 layers of rubber sheets. The uncured fabric reinforced rubber composites were placed into a $23 \times 23 \times 10$ cm$^3$ stainless steel mold and vulcanized at 150 °C for 40 min under 15 MPa.

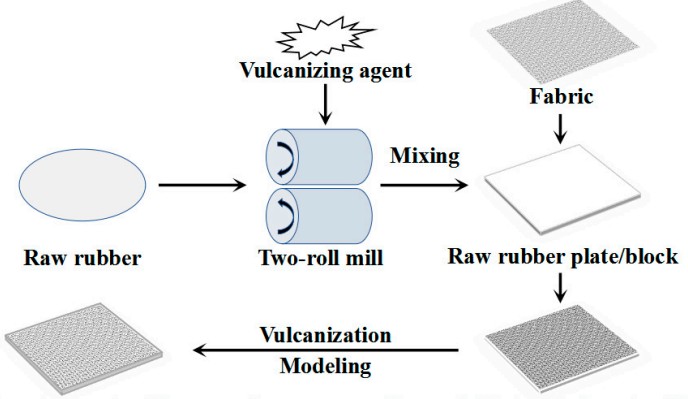

**Figure 1.** The preparation schematic diagram of fabric/rubber composites.

*2.5. Characterizations*

Fourier transform infrared spectroscopy (FTIR, INVENIO S, BRUKER, U.S.A), equipped with an attenuated total reflectance (ATR) accessory containing a diamond crystal, was used to analyze the surface of the fabrics before and after treatment.

The tensile properties of the PET cords were carried out by using a universal testing machine (UTM5504, SUNS, China) at a crosshead speed of 10 cm/min, according to GB/T 3362-2005. The gauge length of the tested fiber was 25.

The morphological surface of the PET cords and the interfacial fracture surface of the composites were examined by scanning electron microscopy (GeminiSEM 360, Carl Zeiss, Germany) with an accelerating voltage of 30 kV.

The peeling strength between the fabrics and the rubber was characterized under ASTM D413-98, at a peeling speed of 50 mm·min$^{-1}$, at room temperature. Strip samples, of length 250 mm and width 25 mm, were used.

A TGA instrument (TGA/DSC1/1600, Mettler, Switzerland) was applied to characterize the grafting content coated on the PET fibers. Ground 5–10 mg specimens from the fibers were used. In order to avoid oxidation of fibers, a constant N$_2$ of 50 mL/min was applied. A heating rate of 20 °C/min was applied to increase the temperature to 700 °C and then maintained at this temperature for 60 min.

The curing characteristics of the rubber composites were tested by a moving die rheometer (MDR3000Basic, Germany), in accordance with ISO 6502:2018 at 160 °C. Then, the curing rate index (CRI) was calculated from Formula (1):

$$\text{CRI} = 100/(t_{90} - t_{10}) \tag{1}$$

in which $t_{90}$ was the positive curing time, and $t_{10}$ was the scorch time.

## 3. Results and Discussion

*3.1. Fiber Strength after Alkali Treatment and Grafting KH550*

In order to analyze the influence of alkali treatment and grafting coupling agent on the strength of the fibers an analysis of the multifilament tensile strength of the PET fibers was performed. Figure 2 shows the relationship between treatment process modes and tensile strength. It was revealed that the tensile strength of fibers decreased with alkali treatment, while it increased when KH550 was grafted on the fibers. When the fabrics were immersed in alkali solution, ethylene glycol and sodium terephthalate were generated

under the hydrolysis of the surface of the fabric [11,12], leaving the OH, -COONa and -COOH groups on the fabric surface. The reaction process is shown in Figure 3. Compared to the untreated fiber, the tensile strength was decreased by the NaOH solution treatment, indicating some damage of fibers and causing a loss of tensile strength. From our results, shown in Figure 2, the tensile strength of fibers inversely recovered a little after KH550 treatment. Since the grooves and holes etched by the alkali solution could be embedded by the polymer molecules, they could be partly repaired (as shown in Figure 4), reducing the probability of stress concentration and avoiding fiber breakage.

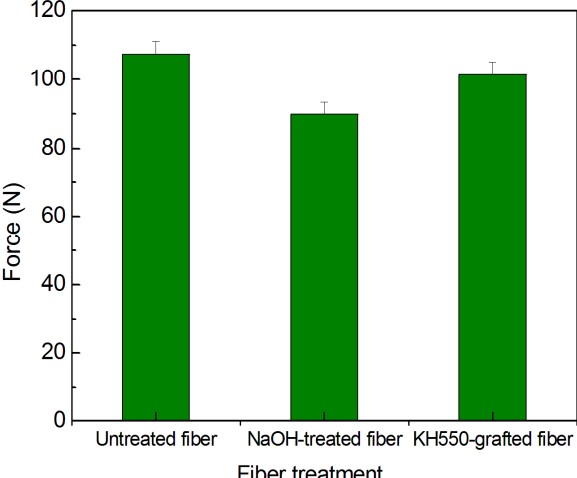

**Figure 2.** Effect of process mode on tensile strength of fibers.

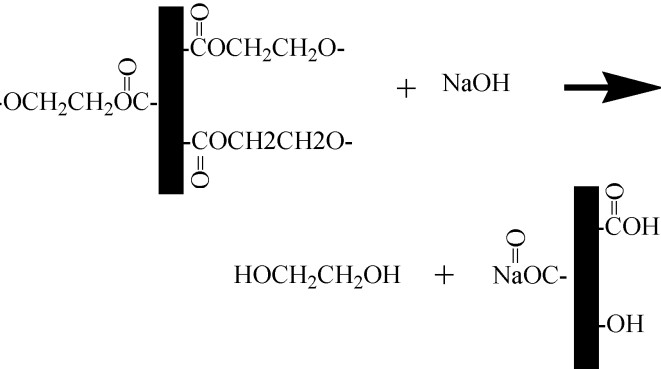

**Figure 3.** Hydrolysis of fiber with NaOH solution.

### 3.2. Surface Modification Analysis of the Fabric

To understand the tendency of fiber strength (Figure 2), SEM was employed to characterize the surface morphology of untreated and treated fabric, shown in Figure 4. From Figure 4a,b, it can be seen that the original fabric surface was obviously neat and smooth. A few aggregates were on the pristine fabric surface, which may have been remaining dust or sodium chloride particles from the PET fabric production [13]. After the fabric was treated with alkali solution, an extremely rough surface and cracks appeared because of the strong active hydrolysis reaction of the PET fabric surface (etch effect), as shown in Figure 4c,d. It can be seen from Figure 4e that a white layer of KH550 coupling agent was coated on the fabric, and it was easy to presume that some KH550 molecules could fill the defects (large holes and grooves) of the fabric (Figure 4f), which was the reason for the enhanced strength, compared with that of the NaOH-treated fabric (Section 3.1).

The chemical structure of KH550 is shown in Figure 5a, and it hydrolyzed easily (Figure 5b). When the alkali-treated fabric was put into the KH550 solution under ultrasonic, the sodium terephthalate on the surface formed into terephthalic acid in aqueous solution,

which reacted with the hydroxyl silane groups (hydrolysate of KH550) and formed water molecules (Figure 5c). Thus, the KH550 was grafted on the fabric surface and the strength was recovered after KH550 treatment.

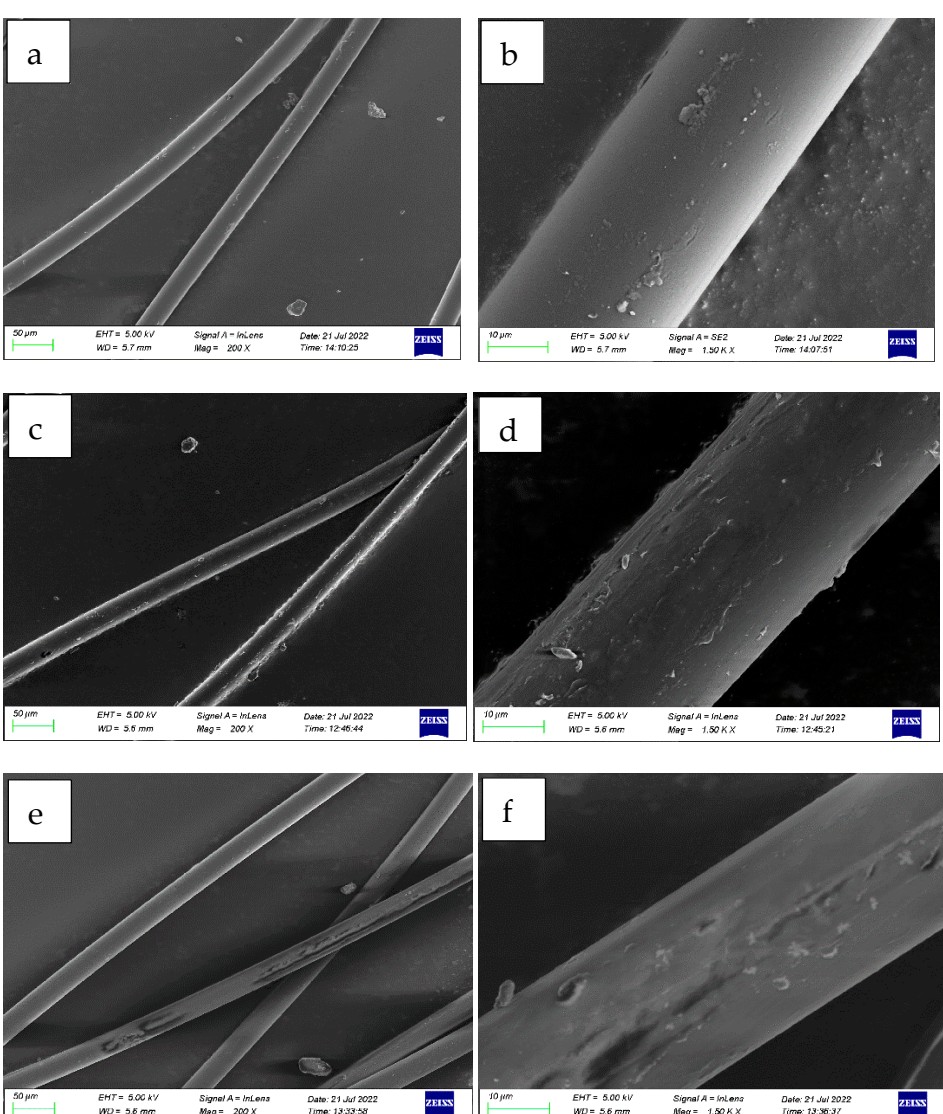

**Figure 4.** SEM images of fabric samples. (**a**,**b**) original fabric; (**c**,**d**) NaOH-treated fabric, (**e**,**f**) KH550-grafted fabric.

FTIR of the fabrics treated by various process modes are shown in Figure 6. The absorption peaks located at 1233 and 1013 cm$^{-1}$ were due to the stretching vibration of the C–O–C group, and they were characteristic peaks of ester groups. The peaks at 1712 cm$^{-1}$ and 2969 cm$^{-1}$ were attributed to stretching vibration, which was caused by groups of C=O and –CH$_2$–, respectively. The –CH– in the benzene ring bending vibration peaked at 745 cm$^{-1}$ [14,15]. The C-O of the polyester fabric changed from 1241 cm$^{-1}$ to 1235 cm$^{-1}$ after the alkali and KH550 coupling agent were added, shown in Figure 6b. It may have been the result of the hydrolysis of ester groups and grafting coupling agent treatment [10].

Figure 7 shows the TGA analysis of the pristine fabric, NaOH-treated fabric and KH550-grafted fabric. The residue rate of the pristine fabric was 14.1 wt%. In this temperature range (room temperature to 700 °C), the PET fabric showed large decomposition. After NaOH solution treatment, the residue rate of the NaOH-treated fabric was 11.1 wt%, because both severe surface hydrolysis of the NaOH-treated fabric and the resulting -OH and -COOH groups decomposed easily. The residue rate of 13.2 wt% was obtained after KH550

treatment, and the grafting rate of KH550 on the fabric surface was 2.1 wt%. This also confirmed that the KH550 molecules had been successfully grafted onto the fabric surface.

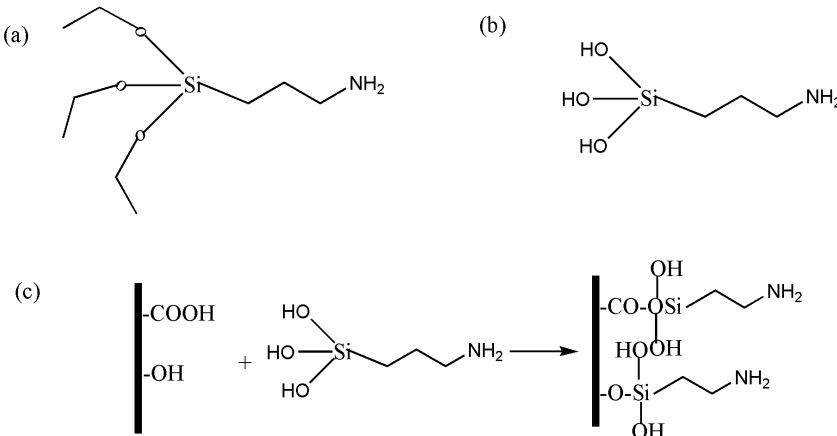

**Figure 5.** (**a**) chemical structures of KH550; (**b**) Schematic illustration of the hydrolysis of KH550 and (**c**) the reaction of KH550 with fabrics.

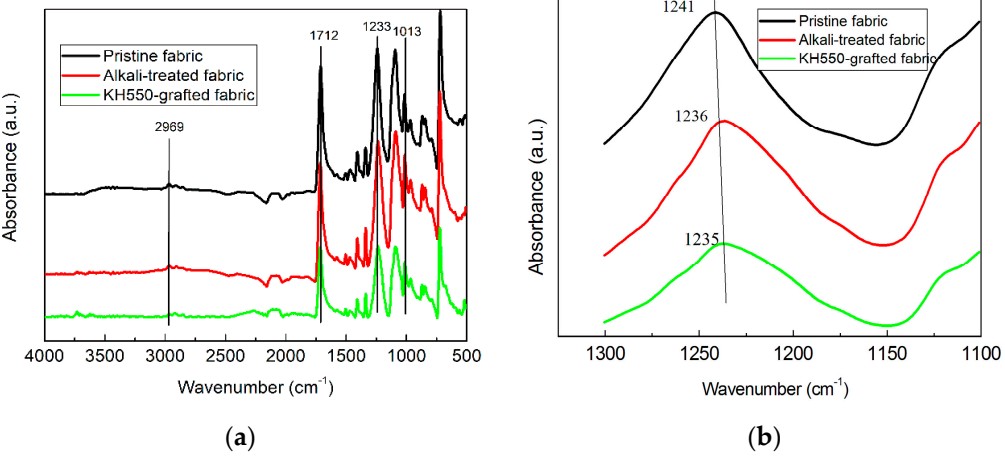

**Figure 6.** FTIR spectrum analysis of the fabric modified by different process modes. (**a**) wide scan spectra of FTIR; (**b**) FTIR spectrum with wavenumbers from 1300 cm$^{-1}$ to 1100 cm$^{-1}$.

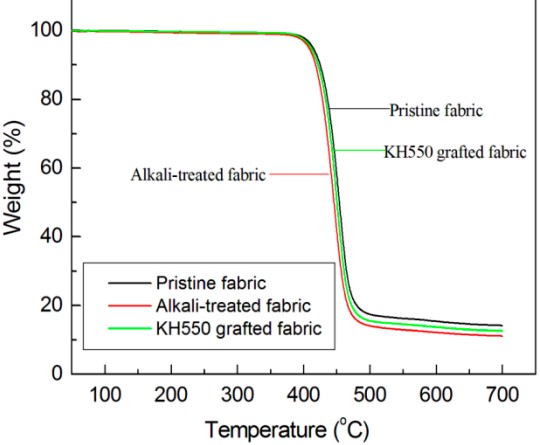

**Figure 7.** Weight loss as a function of time for pristine fabric, NaOH-treated fabric, KH550-grafted fabric.

### 3.3. Vulcanization Characteristics of Fabric/Rubber Composite

Three steps were applied to divide the curing process of the rubber composite: induction stage, fast reaction stage and flat reaction stage [16]. The $t_{10}$ was the scorch time, also called the induction stage, in which time the sample gradually heated from the outside to the inside. However, the rubber could not be vulcanized in this stage and the rubber had good fluidity because of the shear thinning effect and the large heat. The $t_{90}$ was the time when 90% of the maximum torque was achieved and the rubber was rapidly cross-linked. Lots of free radicals were generated from the vulcanizing agent and the torque also increased rapidly under this state.

Figure 8 shows the curing characteristics of the fabric/rubber composite, and the vulcanization parameters are listed in Table 1. As shown in Figure 8 and Table 1, the vulcanization curves of the fabric composites were different from those of the pure rubber. The CRI of pure rubber was 17.95 $min^{-1}$, and the CRIs of the fabric samples were lower. The reason for the decrease of the vulcanization rate may have been the diluting effect of fabric on the vulcanizing agent. Obviously, the fabric and the coupling agent could postpone the vulcanization of rubber.

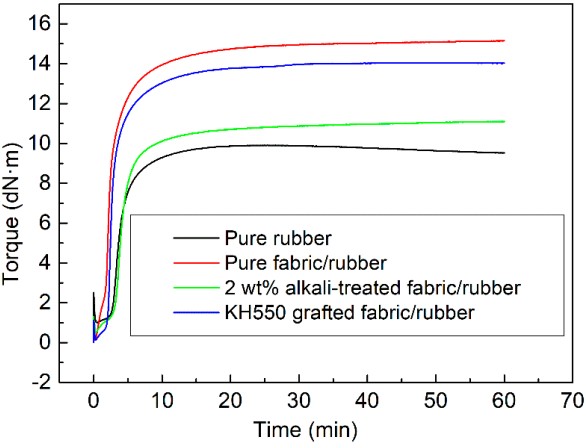

**Figure 8.** Vulcanization curves of samples (Pure rubber, Pure fabric/rubber, 2 wt% alkali-treated fabric/rubber, KH550 grafted fabric/rubber).

**Table 1.** Vulcanization parameters of samples (Pure rubber, Pure fabric/rubber, alkali-treated fabric/rubber, KH550 grafted fabric/rubber).

| Curing Characteristics | Pure Rubber | Original Polyester/Rubber | Alkali-Treated Fabric/Rubber | KH550 Grafted Fabric/Rubber |
|---|---|---|---|---|
| $M_H$ (dN·m) | 9.91 | 15.16 | 11.11 | 14.05 |
| $M_L$ (dN·m) | 1.01 | 0.12 | 0.50 | 0.12 |
| $t_{10}$ (min) | 2.81 | 1.07 | 2.88 | 2.08 |
| $t_{50}$ (min) | 3.62 | 2.33 | 4.02 | 2.66 |
| $t_{90}$ (min) | 8.38 | 8.43 | 9.51 | 8.04 |
| CRI ($min^{-1}$) | 17.95 | 13.59 | 15.08 | 16.78 |

### 3.4. Adhesion Behaviors of the Composites with Alkali-Treated Fabric and KH550 Grafted Fabric

The interfacial adhesion of the fabric/rubber composites was tested by the peel strength. The peel strength of the fabrics under different treatments are shown in Figure 9a. The peel strength of the KH550 grafted fabric/rubber composites was higher than the others when the fabrics were modified by the KH550, which caused an increase of 32% compared with that of the original fabric composites. The peel curves of fabrics treated with different fabric surface treatment are shown in Figure 9b. It was also obvious that the KH550 grafted fabric treatment could effectively improve the interfacial bonding. In addition, the curves of the alkali-treated fabric/rubber composites had greater modulus

and more fluctuations than the other peeling curves. The interfacial adhesion and the fabric shrinkage deformation could affect the peel strength [17]. Obviously, the alkali-treated fabric could be easily deformed because the alkali-treated fabric was damaged to some extent.

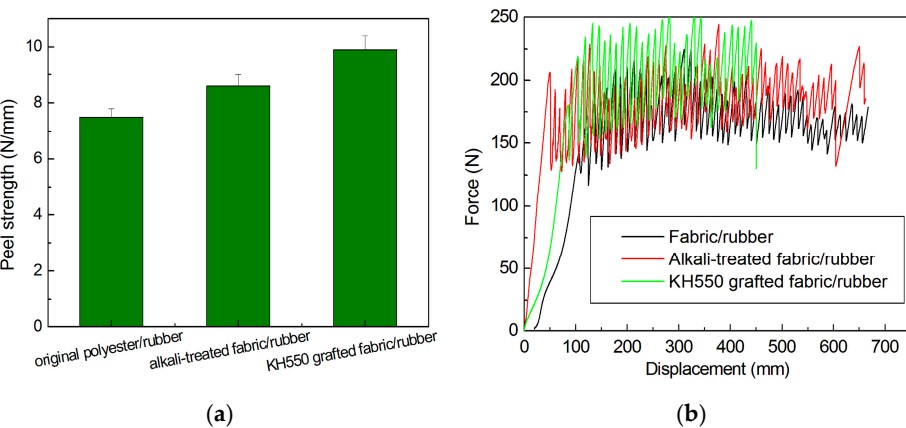

**Figure 9.** (**a**) Peel strength; (**b**) peel curves of fabric/rubber under different fabric surface treatments.

The macro-photographs of the interfacial fracture surface of the fabric composites alkali-treated and with KH550 grafted after the 180° peel test are presented in Figure 10. The fabric surface was bright and the rubber was black (Figure 10a), which indicated that the interfacial adhesion was low. A wholly dark surface of fabric compared with that of the original fabric meant that the surface had been partly covered and a rubber layer had formed, indicating that the interfacial adhesion was larger than that of the original fabric/rubber composites (Figure 10b). A partly cohesive rubber failure occurred when the fabric was treated by alkali solution. As shown in Figure 10c, a large amount of rubber adhered to the fabrics and the rubber layer also had a lot of holes (red circles). These characteristics indicated a complete failure in the bulk of the fabric/rubber composites, not only the interfacial failure. Hence, KH550-grafted fabric/rubber composites had the strongest interfaces to resist stripping.

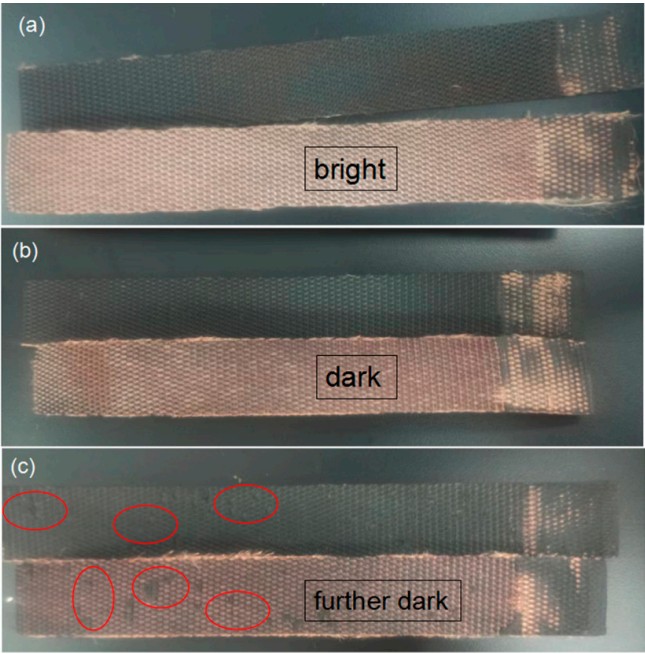

**Figure 10.** The photograph of interfacial fracture surface of fabric/rubber composites after the peel test. (**a**) original fabric/rubber, (**b**) 2 wt% alkali-treated fabric/rubber, (**c**) KH550-grafted fabric/rubber.

## 4. Conclusions

In this paper, alkaline and coupling agent treatment for fabric was developed to improve the problem of the poor adhesion between the fabric and rubber. The tensile strength of the fibers did not largely decrease after alkaline and coupling agent treatment. The tensile strength of fibers with coupling agent treatment was inversely partly recovered, compared with alkaline treatment. ATR-FTIR and TGA showed that the KH550 could be grafted on the fabric surface and the strength partly recovered after KH550 treatment. The peel strength reached 9.8 N/mm when the fabric was modified by KH550, which was an increase of 32%, compared with that of the original fabric composites. A complete failure in the bulk of the KH550-grafted fabric/rubber composites was found, and this could greatly improve the stripping resistance of fabric/rubber composites and expand their industrial applications.

**Supplementary Materials:** The following supporting information can be downloaded at: https://www.mdpi.com/article/10.3390/coatings12091344/s1, Table S1: The formula of traditional RFL dipping solution. Table S2: The Rubber Compound Composition.

**Author Contributions:** Conceptualization, H.H., Q.Z. and P.W.; methodology, Z.Y.; software, Z.S.; validation, W.Y.; writing—original draft preparation, H.H.; writing—review and editing, H.H.; supervision, F.L. and F.Z.; project administration, Q.Z., D.Z. and S.L. All authors have read and agreed to the published version of the manuscript.

**Funding:** The authors wish to thank the financial support from Natural Science Foundation of Shanxi Province, China (No. 202103021224102), supported by the Fund for Shanxi "1331 Project", "New polymer functional materials" industry-university-research innovation platform (DC2100000856), the Key Research and Development (R&D) Projects of Shanxi Province (No. 202102040201011).

**Institutional Review Board Statement:** Not applicable.

**Informed Consent Statement:** Not applicable.

**Data Availability Statement:** Not applicable.

**Conflicts of Interest:** The authors declare no conflict of interest.

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
