# Peer review of "A Facile Way to Modify Polyester Fabric to Enhance the Adhesion Behavior to Rubber"

_coatings, doi:10.3390/coatings12091344_

Round 1

Reviewer 1 Report

Comments

Comments to authors

1. mentioned the chemical composition of materials in section 2.1.

2. It is mandatory to provide the refereve in section 2.3, for sentence "Formaldehyde and resorcinol were mixed in water and then sodium hydroxide as a 83 catalyst was added to the mixture and then the mixture was matured for 6 h at 25 oC, 84 forming matured RF resin"

3. Mentioned the size of tensile test specimens in section 2.5.

4. The tensile strength of materials must shows in MPa in Figure 2. Modify it.

5 SEM image Figure 4 (c2) shows more roughness and cracks in the comparison of Figure 4 (b2), which is in contradiction with the author's statements. Justify the same, because more roughness should result in a decrease in tensile strength.

6. Authors put the wrong figure numbers in section 3.2. correct it.

7.Mentioned some concrete conclusions in section 4.

Reviewer 2 Report

Line 57-58: Results presented in introduction section.

Line 141-143: Ppoint made here is repeated in lines 155-158

Infrared Analysis Section: Peak shift cannot be presented as evidence of functionalization without at least presenting multiple measurements that support this finding? What is the interaction depth of the ATR probe with the sample surface? How many spectra were recorded for each type of surface? Were any other spectral analysis methods applied to analyse the results such as subtraction, derivitization or multivariate analysis techniques?

Thermal Analysis: What is the residue rate? Is this the remaining residue? What variation in the residue was observed for replicates of the same sample? Does a 2 % mass difference after surface functionalisation sound reasonable? 

Reviewer 3 Report

Hongwei He. et al., elaborated the manuscript titled: ”A facile way to modify polyester fabric to enhance the adhesion behavior to rubber”, while the developed facile way to modify PET fabric was presented. Consistent data regarding surface chemical compositions of the modified fabrics and most important the adhesion behavior was analyzed. The primary aim of the work is very interesting, authors investigating alternatives to the conventional toxic process of RFL dipping solution. Generally, the structure of the manuscript is well written, however some details are missing, thus I recommend a minor revision before publishing the manuscript. Authors should address the following punctual observations:

-          The abstract section is well written, however the introduction is too summarized; Authors should consider citing other groups involved in the improving the polyester fabric; What is the novelty of the proposed research and what is the current status of the investigated topic;

-          Authors state in line 57: “the adhesion of PET fabrics to rubber can be reached to 9.8 N/mm, which provides a useful reference for the engineering application.”. The statement should include a reference. Also, in the manuscript, authors should consider mentioning how they are correlating their results with the mentioned value;

-          Section 2.1 could be summarized as a table, just as a recommendation;

-          In line 85: RF acronym should be a described;

-          The components of traditional RFL dipping solution are given in Table S1…..The composition of the rubber compound as shown in Table S2 was used in our study.” Where are those tables in the manuscript?;

-          Add the starting and the finishing points in figure 1 in order for the reader to better understand;

-          What is the standard deviation for the tensile strength values presented in figure 2? Can authors compare those values to the literature?

-          Authors mention in line 152: “and a few particles on the surface of the original fabric may be dust or sodium chloride particles not cleaned up during the preparation process [13]”; It was there an option to further improve the preparation process? Please comment;

-          Figure 4, increase the scale bar; To some readers, the WD and magnification could be relevant, thus please make them visible;

-          Correct the figure numbers in line 155: “as shown in Fig. 7(b1, b2). It can be seen  from Fig. 7(c1)”; “figure in Fig. 4(b), the characteristic peak of C-O of the polyester fabric changes”;

-          Please insert the corresponding lines in the FTIR spectrum (figure 6.a);

-          How the error bars were calculated in figure 9.a;

-          Authors state: “The reason for the decrease of the vulcanization rate maybe the diluting effect of fabric on vulcanizing agent.”; It would be useful to support this supposition by citing a paper; comparing with other findings;

Author Response

Please see accessory

Round 2

Reviewer 1 Report

The authors' responses are satisfactory. I have recommended this manuscript for acceptance.

Author Response

Thank you.

Reviewer 2 Report

The article is not ready for publication in its current form and requires more analysis in terms of the spectroscopic and thermogravimetric data.

Author Response

First, we want to say that we are so sorry for limits of levels of our knowledge, and our responses may not be full and perfect to the reviewer. Second, we have also consulted some researchers for infrared spectroscopic and thermogravimetric data, including equipment manufacturers, and the statements in the paper are acceptable in general, although not perfect. We will elaborate our Response from the following three points.

Please check the accessory. Thank you
